# EMI: Exploration with Mutual Information Maximizing State and Action Embeddings

## Abstract

Policy optimization struggles when the reward feedback signal is very sparse and essentially becomes a random search algorithm until the agent accidentally stumbles upon a rewarding or the goal state. Recent works utilize intrinsic motivation to guide the exploration via generative models, predictive forward models, or more ad-hoc measures of surprise. We propose EMI, which is an exploration method that constructs embedding representation of states and actions that does not rely on generative decoding of the full observation but extracts predictive signals that can be used to guide exploration based on forward prediction in the representation space. Our experiments show the state of the art performance on challenging locomotion task with continuous control and on image-based exploration tasks with discrete actions on Atari.

## 1 Introduction

The central task in reinforcement learning is to learn policies that would maximize the total reward received from interacting with the unknown environment. Although recent methods have demonstrated to solve a range of complex tasks (Mnih et al., 2015; Schulman et al., 2015; 2017), the success of these methods, however, hinges on whether the agent constantly receives the intermediate reward feedback or not. In case of challenging environments with sparse reward signals, these methods struggle to obtain meaningful policies unless the agent luckily stumbles into the rewarding or predefined goal states.

To this end, prior works on exploration generally utilize some kind of intrinsic motivation mechanism to provide a measure of surprise. These measures can be based on density estimation via generative models (Bellemare et al., 2016; Fu et al., 2017; Oh et al., 2015), predictive forward models (Stadie et al., 2015; Houthooft et al., 2016), or more ad-hoc measures that aim to approximate surprise (Pathak et al., 2017). Methods based on predictive forward models and generative models must model the distribution over state observations, which can make them difficult to scale to complex, high-dimensional observation spaces, while models that eschew direct forward predictive or density estimation rely on heuristic measures of surprise that may not transfer effectively to a wide range of tasks.

Our aim in this work is to devise a method for exploration that does not require a direct generation of high-dimensional state observations, while still retaining the benefits of being able to measure surprise based on the forward prediction. If exploration is performed by seeking out states that maximize surprise, the problem, in essence, is in measuring surprise, which requires a representation where functionally similar states are close together, and functionally distinct states are far apart.

In this paper, we propose to learn compact representations for both the states ($\phi$) and actions ($\psi$) simultaneously satisfying the following criteria: First, given the representations of state and the corresponding next state, the uncertainty of the representation of the corresponding action should be minimal. Second, given the representations of the state and the corresponding action, the uncertainty of the representation of the corresponding next state should also be minimal. Third, the action embedding representation ($\psi$) should seamlessly support both the continuous and discrete actions. Finally, we impose the linear dynamics model in the representation space which can also explain the rare irreducible error under the dynamics model. Given the representation, we guide the exploration by measuring surprise based on forward prediction and relative increase in diversity in the embedding representation space. Figure 1 illustrates an example visualization of our learned state embedding representations ($\phi$) and sample trajectories in the representation space in Montezuma's Revenge.

We present two main technical contributions that make this into a practical exploration method. First, we describe how compact state and action representations can be constructed via Donsker & Varadhan (1983) estimation of mutual information without relying on generative decoding of full

observations. Second, we show that imposing linear topology on the learned embedding representation space (such that the transitions are linear), thereby offloading most of the modeling burden onto the embedding function itself, provides an essential informative measure of surprise when visiting novel states.

For the experiments, we show that we can use our representations on a range of complex image-based tasks and robotic locomotion tasks with continuous actions. We report significantly improved results compared to recent intrinsic motivation based exploration methods (Fu et al., 2017; Pathak et al., 2017) on several challenging Atari tasks and robotic locomotion tasks with sparse rewards.

Figure 1: Visualization of sample trajectories in our learned embedding space.

## 2 RELATED WORKS

Our work is related to the following strands of active research:

**Unsupervised representation learning via mutual information estimation**  Recent literature on unsupervised representation learning generally focus on extracting latent representation maximizing approximate lower bound on the mutual information between the code and the data. In the context of generative adversarial networks (Goodfellow et al., 2014), Chen et al. (2016); Belghazi et al. (2018) aims at maximizing the approximation of mutual information between the latent code and the raw data. Belghazi et al. (2018) estimates the mutual information with neural network via Donsker & Varadhan (1983) estimation to learn better generative model. Hjelm et al. (2018) builds on the idea and trains a decoder-free encoding representation maximizing the mutual information between the input image and the representation. Furthermore, the method uses $f$-divergence (Nowozin et al., 2016) estimation of Jensen-Shannon divergence rather than the KL divergence to estimate the mutual information for better numerical stability. Oord et al. (2018) estimates mutual information via autoregressive model and makes predictions on local patches in an image. Thomas et al. (2017) aims to learn the representations that maximize the causal relationship between the distributed policies and the representation of changes in the state.

**Exploration with intrinsic motivation**  Prior works on exploration mostly employ intrinsic motivation to estimate the measure of novelty or surprisal to guide the exploration. Bellemare et al. (2016) utilize density estimation via CTS (Bellemare et al., 2014) generative model and derive pseudo-counts as the intrinsic motivation. Fu et al. (2017) avoids building explicit density models by training K-exemplar models that distinguish a state from all other observed states. Some methods train predictive forward models (Stadie et al., 2015; Houthooft et al., 2016; Oh et al., 2015) and estimate the prediction error as the intrinsic motivation. Oh et al. (2015) employs generative decoding of the full observation via recursive autoencoders and thus can be challenging to scale for high dimensional observations. VIME (Houthooft et al. (2016)) approximates the environment dynamics, uses the information gain of the learned dynamics model as intrinsic rewards, and showed encouraging results on robotic locomotion problems. However, the method needs to update the dynamics model per each observation and is unlikely to be scalable for complex tasks with high dimensional states such as Atari games.

Other approaches utilize more ad-hoc measures (Pathak et al., 2017; Tang et al., 2017) that aim to approximate surprise. ICM (Pathak et al. (2017)) transforms the high dimensional states to feature space and imposes cross entropy and Euclidean loss so the action and the feature of the next state are predictable. However, ICM does not utilize mutual information like VIME to directly measure the uncertainty and is limited to discrete actions. Our method (EMI) is also reminiscent of (Kohonen & Somervuo, 1998) in a sense that we seek to construct a decoder-free latent space from the high dimensional observation data with a topology in the latent space. In contrast to the prior works on exploration, we seek to construct the representation under linear topology and does not require decoding the full observation but seek to encode the essential predictive signal that can be used for guiding the exploration.

## 3 PRELIMINARIES

We consider a Markov decision process defined by the tuple $(\mathcal{S}, \mathcal{A}, P, r, \gamma)$, where $\mathcal{S}$ is the set of states, $\mathcal{A}$ is the set of actions, $P : \mathcal{S} \times \mathcal{A} \times \mathcal{S} \to \mathbb{R}_+$ is the environment transition distribution, $r : \mathcal{S} \to \mathbb{R}$ is the reward function, and $\gamma \in (0, 1)$ is the discount factor. Let $\pi$ denote a stochastic policy over actions given states. Denote $\mathbb{P}_0 : \mathcal{S} \to \mathbb{R}_+$ as the distribution of initial state $s_0$. The discounted sum of expected rewards under the policy $\pi$ is defined by

$$\eta(\pi) = \mathbb{E}_\tau \left[ \sum_{t=0} \gamma^t r(s_t) \right], \tag{1}$$

where $\tau = (s_0, a_0, \ldots, a_{T-1}, s_T)$ denotes the trajectory, $s_0 \sim \mathbb{P}_0(s_0), a_t \sim \pi(a_t \mid s_t)$, and $s_{t+1} \sim P(s_{t+1} \mid s_t, a_t)$. The objective in policy based reinforcement learning is to search over the space of parameterized policies (*i.e.* neural network) $\pi_\theta(a \mid s)$ in order to maximize $\eta(\pi_\theta)$.

Also, denote $\mathbb{P}_{SAS'}^\pi$ as the joint probability distribution of singleton experience tuples $(s, a, s')$ starting from $s_0 \sim \mathbb{P}_0(s_0)$ and following the policy $\pi$. Furthermore, define $\mathbb{P}_A^\pi = \int_{\mathcal{S} \times \mathcal{S}'} d\mathbb{P}_{SAS'}^\pi$ as the marginal distribution of actions, $\mathbb{P}_{SS'}^\pi = \int_{\mathcal{A}} d\mathbb{P}_{SAS'}^\pi$ as the marginal distribution of states and the corresponding next states, $\mathbb{P}_{S'}^\pi = \int_{\mathcal{S} \times \mathcal{A}} d\mathbb{P}_{SAS'}^\pi$ as the marginal distribution of the next states, and $\mathbb{P}_{SA}^\pi = \int_{\mathcal{S}'} d\mathbb{P}_{SAS'}^\pi$ as the marginal distribution of states and the actions following the policy $\pi$.

## 4 METHODS

Our goal is to construct the embedding representation of the observation and action (discrete or continuous) for complex dynamical systems that does not rely on generative decoding of the full observation, but still provides a useful predictive signal that can be used for exploration. This requires a representation where functionally similar states are close together, and functionally distinct states are far apart. We approach this objective from maximizing mutual information under several criteria.

### 4.1 MUTUAL INFORMATION MAXIMIZING STATE AND ACTION EMBEDDING REPRESENTATIONS

We first introduce the embedding function of states $\phi_\alpha : \mathcal{S} \to \mathbb{R}^d$ and actions $\psi_\beta : \mathcal{A} \to \mathbb{R}^d$ with parameters $\alpha$ and $\beta$ (*i.e.* neural networks) respectively. We seek to learn the embedding function of states ($\phi_\alpha$) and actions ($\psi_\beta$) satisfying the following two criteria:

1. Given the embedding representation of states and the actions $[\phi_\alpha(s); \psi_\beta(a)]$, the uncertainty of the embedding representation of the corresponding next states $\phi_\alpha(s')$ should be minimal and vice versa.

2. Given the embedding representation of states and the corresponding next states $[\phi_\alpha(s); \phi_\alpha(s')]$, the uncertainty of the embedding representation of the corresponding actions $\psi_\beta(a)$ should also be minimal and vice versa.

Intuitively, the first criterion translates to maximizing the mutual information between $[\phi_\alpha(s); \psi_\beta(a)]$, and $\phi_\alpha(s')$ which we define as $\mathcal{I}_S(\alpha, \beta)$ in Equation (2). And the second criterion translates to maximizing the mutual information between $[\phi_\alpha(s); \phi_\alpha(s')]$ and $\psi_\beta(a)$ defined as $\mathcal{I}_A(\alpha, \beta)$ in Equation (3).

$$\operatorname*{maximize}_{\alpha, \beta} \mathcal{I}_S(\alpha, \beta) := \mathcal{I}([\phi_\alpha(s); \psi_\beta(a)]; \phi_\alpha(s')) = \mathcal{D}_{\mathrm{KL}} \left( \mathbb{P}_{SAS'}^\pi \,\|\, \mathbb{P}_{SA}^\pi \otimes \mathbb{P}_{S'}^\pi \right) \tag{2}$$

$$\operatorname*{maximize}_{\alpha, \beta} \mathcal{I}_A(\alpha, \beta) := \mathcal{I}([\phi_\alpha(s); \phi_\alpha(s')]; \psi_\beta(a)) = \mathcal{D}_{\mathrm{KL}} \left( \mathbb{P}_{SAS'}^\pi \,\|\, \mathbb{P}_{SS'}^\pi \otimes \mathbb{P}_A^\pi \right) \tag{3}$$

Mutual information is not bounded from above and maximizing mutual information is notoriously difficult to compute in high dimensional settings. Motivated by Hjelm et al. (2018); Belghazi et al. (2018), we compute Donsker & Varadhan (1983) lower bound of mutual information. Concretely, Donsker-Varadhan representation is a tight estimator for the mutual information of two random variables $X$ and $Z$, derived as in Equation (4).

$$\mathcal{I}(X; Z) = \mathcal{D}_{\mathrm{KL}}(\mathbb{P}_{XZ} \,\|\, \mathbb{P}_X \otimes \mathbb{P}_Z) \geq \sup_{\omega \in \Omega} \mathbb{E}_{\mathbb{P}_{XZ}} T_\omega(x, z) - \log \mathbb{E}_{\mathbb{P}_X \otimes \mathbb{P}_Z} \exp T_\omega(x, z), \tag{4}$$

where $T_\omega : \mathcal{X} \times \mathcal{Z} \to \mathbb{R}$ is a differentiable transform with parameter $\omega$. Furthermore, for better numerical stability, we utilize a different measure between the joint and marginals than the KL-

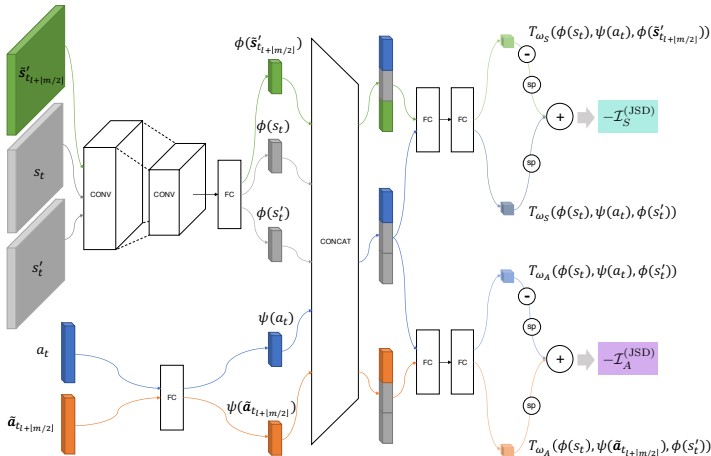

Figure 2: Computational architecture for estimating $\mathcal{I}_S^{(\text{JSD})}$ and $\mathcal{I}_A^{(\text{JSD})}$ for image-based observations.

divergence. In particular, we employ Jensen-Shannon divergence (JSD) (Hjelm et al., 2018) which is bounded both from below and above by 0 and $\log(4)$ [1].

**Theorem 1.** $\mathcal{I}^{(\text{JSD})}(X;Z) \geq \sup_{\omega \in \Omega} \mathbb{E}_{\mathbb{P}_{XZ}} \left[-\text{sp}\left(-T_\omega(x,z)\right)\right] - \mathbb{E}_{\mathbb{P}_X \otimes \mathbb{P}_Z} \left[\text{sp}\left(T_\omega(x,z)\right)\right] + \log(4)$

*Proof.*
$$\begin{aligned}
\mathcal{I}^{(\text{JSD})}(X;Z) &= \mathcal{D}_{\text{JSD}}(\mathbb{P}_{XZ} \parallel \mathbb{P}_X \otimes \mathbb{P}_Z) \\
&\geq \sup_{\omega \in \Omega} \mathbb{E}_{\mathbb{P}_{XZ}} \left[S_\omega(x,z)\right] - \mathbb{E}_{\mathbb{P}_X \otimes \mathbb{P}_Z} \left[\text{JSD}^* \left(S_\omega(x,z)\right)\right] \\
&= \sup_{\omega \in \Omega} \mathbb{E}_{\mathbb{P}_{XZ}} \left[-\text{sp}\left(-T_\omega(x,z)\right)\right] - \mathbb{E}_{\mathbb{P}_X \otimes \mathbb{P}_Z} \left[\text{sp}\left(T_\omega(x,z)\right)\right] + \log(4),
\end{aligned} \qquad (5)$$

where the inequality in the second line holds from the definition of $f$-divergence (Nowozin et al., 2016). In the third line, we substituted $S_\omega(x,z) = \log(2) - \log(1 + \exp(-T_\omega(x,z)))$ and *Fenchel conjugate* of Jensen-Shannon divergence, $\text{JSD}^*(t) = -\log(2 - \exp(t))$. $\qquad \square$

From Theorem 1, we have,

$$\underset{\alpha,\beta}{\text{maximize }} \mathcal{I}_S^{(\text{JSD})}(\alpha,\beta) \geq \underset{\alpha,\beta}{\text{maximize }} \sup_{\omega_S \in \Omega_S} \mathbb{E}_{\mathbb{P}_{SAS'}^\pi} \left[-\text{sp}\left(-T_{\omega_S}(\phi_\alpha(s), \psi_\beta(a), \phi_\alpha(s'))\right)\right] \qquad (6)$$
$$- \mathbb{E}_{\mathbb{P}_{SA}^\pi \otimes \mathbb{P}_{S'}^\pi} \left[\text{sp}\left(T_{\omega_S}(\phi_\alpha(s), \psi_\beta(a), \phi_\alpha(\tilde{s}'))\right)\right] + \log 4,$$

$$\underset{\alpha,\beta}{\text{maximize }} \mathcal{I}_A^{(\text{JSD})}(\alpha,\beta) \geq \underset{\alpha,\beta}{\text{maximize }} \sup_{\omega_A \in \Omega_A} \mathbb{E}_{\mathbb{P}_{SAS'}^\pi} \left[-\text{sp}\left(-T_{\omega_A}(\phi_\alpha(s), \psi_\beta(a), \phi_\alpha(s'))\right)\right] \qquad (7)$$
$$- \mathbb{E}_{\mathbb{P}_{SS'}^\pi \otimes \mathbb{P}_A^\pi} \left[\text{sp}\left(T_{\omega_A}(\phi_\alpha(s), \psi_\beta(\tilde{a}), \phi_\alpha(s'))\right)\right] + \log 4,$$

where $\text{sp}(z) = \log(1 + \exp z)$. The expectations in Equation (6) and Equation (7) are approximated using the empirical samples trajectories $\tau$. Note, the samples $\tilde{s}' \sim \mathbb{P}_{S'}^\pi$ and $\tilde{a} \sim \mathbb{P}_A^\pi$ from the marginals are obtained by dropping $(s, a)$ and $(s, s')$ in samples $(s, a, \tilde{s}')$ and $(s, \tilde{a}, s')$ from $\mathbb{P}_{SAS'}^\pi$. Figure 2 illustrates the computational architecture for estimating the lower bounds on $\mathcal{I}_S$ and $\mathcal{I}_A$.

## 4.2 EMBEDDING LINEAR DYNAMICS MODEL UNDER SPARSE NOISE

Since the embedding representation space is learned, it is natural to impose a topology on it (Kohonen, 1983). In EMI, we impose a simple and convenient topology where transitions are linear since this spares us from having to also represent a complex dynamical model. This allows us to offload most of the modeling burden onto the embedding function itself, which in turn provides us with a useful and informative measure of surprise when visiting novel states. Once the embedding representations are learned, this linear dynamics model allows us to measure *surprise* in terms of the residual error under the model or measure *diversity* in terms of the similarity in the embedding space. Section 5 discusses the intrinsic reward computation procedure in more detail.

---

[1] In Nowozin et al. (2016), the authors actually derived the lower bound of $D_{JSD} = D_{KL}(P\|M) + D(Q\|M)$, instead of $D_{JSD} = \frac{1}{2}(D_{KL}(P\|M) + D_{KL}(Q\|M))$, where $M = \frac{1}{2}(P + Q)$.

Concretely, we seek to learn the representation of states $\phi(s)$ and the actions $\psi(a)$ such that the representation of the corresponding next state $\phi(s')$ follow linear dynamics *i.e.* $\phi(s') = \phi(s) + \psi(a)$. Intuitively, we would like the nonlinear aspects of the dynamics to be offloaded to the neural networks $\phi(\cdot), \psi(\cdot)$ so that in the $\mathbb{R}^d$ embedding space, the dynamics become linear. Regardless of the expressivity of the neural networks, however, there always exists irreducible error under the linear dynamic model. For example, the state transition which leads the agent from one room to another in Atari environments (*i.e.* Venture, Montezuma's revenge, *etc.*) or the transition leading the agent in the same position under certain actions (*i.e.* Agent bumping into a wall when navigating a maze environment) would be extremely challenging to explain under the linear dynamics model.

To this end, we introduce the error model $S_\gamma : \mathcal{S} \times \mathcal{A} \to \mathbb{R}^d$, which is another neural network taking the state and action as input, estimating the irreducible error under the linear model. Motivated by the work of Candès et al. (2011), we seek to minimize for the sparsity of the term so that the error term contributes only on rare unexplainable occasions. Equation (8) shows the embedding learning problem under linear dynamics with sparse errors.

$$\underset{\alpha,\beta,\gamma}{\text{minimize}} \ \underbrace{\|S_\gamma\|_{2,0}}_{\text{error sparsity}}$$

$$\text{subject to} \quad \underbrace{\Phi'_\alpha = \Phi_\alpha + \Psi_\beta + S_\gamma}_{\text{embedding linear dynamics}}, \tag{8}$$

where we used the matrix notation for compactness. $\Phi_\alpha, \Psi_\beta, S_\gamma$ denotes the matrices of respective embedding representations stacked columns wise. Relaxing the $\ell_0$ norm with $\ell_1$ norm, Equation (9) shows our final learning objective.

$$\underset{\alpha,\beta,\gamma}{\text{minimize}} \ \|\Phi'_\alpha - (\Phi_\alpha + \Psi_\beta + S_\gamma)\|_{2,1} + \lambda_{\text{sparsity}}\|S_\gamma\|_{2,1} + \lambda_{\text{unitKL}}\mathcal{D}_{\text{KL}}(\mathbb{P}^\pi_\psi \parallel \mathcal{N}(0,I)) \tag{9}$$

$$+ \lambda_{\text{info}} \inf_{\omega_S \in \Omega_S} \mathbb{E}_{\mathbb{P}^\pi_{SAS'}} \text{sp}\left(-T_{\omega_S}(\phi_\alpha(s), \psi_\beta(a), \phi_\alpha(s'))\right) + \mathbb{E}_{\mathbb{P}^\pi_{SA} \otimes \mathbb{P}^\pi_{S'}} \text{sp}\left(T_{\omega_S}(\phi_\alpha(s), \psi_\beta(a), \phi_\alpha(\tilde{s}'))\right)$$

$$+ \lambda_{\text{info}} \inf_{\omega_A \in \Omega_A} \mathbb{E}_{\mathbb{P}^\pi_{SAS'}} \text{sp}\left(-T_{\omega_A}(\phi_\alpha(s), \psi_\beta(a), \phi_\alpha(s'))\right) + \mathbb{E}_{\mathbb{P}^\pi_{SS'} \otimes \mathbb{P}^\pi_A} \text{sp}\left(T_{\omega_A}(\phi_\alpha(s), \psi_\beta(\tilde{a}), \phi_\alpha(s'))\right)$$

$\lambda_{\text{info}}, \lambda_{\text{sparsity}}, \lambda_{\text{unitKL}}$ are hyper-parameters which control the relative contributions of the linear dynamics error and the sparsity. In practice, we found the optimization process to be more stable when we further regularize the distribution of action embedding representation to follow a predefined prior distribution. Concretely, we regularize the action embedding distribution to follow a standard normal distribution via $\mathcal{D}_{\text{KL}}(\mathbb{P}^\pi_\psi \parallel \mathcal{N}(0,I))$ similar to VAEs Kingma & Welling (2013). Intuitively, this has the effect of grounding the distribution of action embedding representation (and consequently the state embedding representation) across different iterations of the learning process. [2]

## 5 INTRINSIC REWARD AUGMENTATION

We consider two different formulations of computing the intrinsic reward. First, we consider a relative difference in the novelty of state representations based on the distance in the embedding representation space similar to Oh et al. (2015) as shown in Equation (10). The relative difference makes sure the intrinsic reward diminishes to zero (Ng et al., 1999) once the agent has sufficiently explored the state space. Also, we consider a formulation based on the prediction error under the linear dynamics model as shown in Equation (11). This formulation incorporates the sparse error term and makes sure we differentiate the irreducible error that does not contribute as the novelty.

$$r_d(s_t, a_t, s'_t) = g(s_t) - g(s'_t), \quad \text{where} \quad g(s) = \frac{1}{n}\sum_{i=1}^n \exp\left(-\frac{\|\phi(s) - \phi(s_i)\|^2}{2\sigma^2}\right) \tag{10}$$

$$r_e(s_t, a_t, s'_t) = \|\phi(s_t) + \psi(a_t) + S(s_t, a_t) - \phi(s'_t)\|^2 \tag{11}$$

Note the relative diversity term should be computed *after* the representations are updated based on the samples from the latest trajectories while the prediction error term should be computed *before* the update. Algorithm 1 shows the complete learning procedure in detail.

---

[2]Note, regularizing the distribution of state embeddings instead renders the optimization process much more unstable. This is because the distribution of states are much more likely to be skewed than the distribution of actions, especially during the initial stage of optimization, so the Gaussian approximation becomes much less accurate in contrast to the distribution of actions.

---

**Algorithm 1** Exploration with mutual information state and action embeddings (EMI)

---

**initialize** $\alpha, \beta, \gamma, \omega_A, \omega_S$
  **for** $i = 1, \ldots,$ MAXITER **do**
    Collect samples $\{(s_t, a_t, s'_t)\}_{t=1}^n$ with policy $\pi_\theta$
    Compute residual error intrinsic rewards $\{r_e(s_t, a_t, s'_t)\}_{t=1}^n$ following Equation (11)
    **for** $j = 1, \ldots,$ OPTITER **do**
      **for** $k = 1, \ldots, \lfloor \frac{n}{m} \rfloor$ **do**
        Sample a minibatch $\{(s_{t_l}, a_{t_l}, s'_{t_l})\}_{l=1}^m$
        Compute $\left\{ T_{\omega_A}\left(\phi(s_{t_l}), \psi(a_{t_l}), \phi(s'_{t_l})\right) \right\}_{l=1}^{\lfloor \frac{m}{2} \rfloor}$ and $\left\{ T_{\omega_A}\left(\phi(s_{t_l}), \psi\left(\tilde{a}_{t_{l+\lfloor \frac{m}{2} \rfloor}}\right), \phi(s'_{t_l})\right) \right\}_{l=1}^{\lfloor \frac{m}{2} \rfloor}$
        to derive the lower bound on $\mathcal{I}_A^{\text{(JSD)}}(\alpha, \beta)$ in Equation (7)
        Compute $\left\{ T_{\omega_S}\left(\phi(s_{t_l}), \psi(a_{t_l}), \phi(s'_{t_l})\right) \right\}_{l=1}^{\lfloor \frac{m}{2} \rfloor}$ and $\left\{ T_{\omega_S}\left(\phi(s_{t_l}), \psi(a_{t_l}), \phi\left(\tilde{s}'_{t_{l+\lfloor \frac{m}{2} \rfloor}}\right)\right) \right\}_{l=1}^{\lfloor \frac{m}{2} \rfloor}$
        to derive the lower bound on $\mathcal{I}_S^{\text{(JSD)}}(\alpha, \beta)$ in Equation (6)
        Update $\alpha, \beta, \gamma, \omega_A, \omega_S$ using the Adam (Kingma & Ba, 2015) update rule to minimize Equation (9)
      **end for**
    **end for**
    Compute diversity intrinsic rewards $\{r_d(s_t, a_t, s'_t)\}_{t=1}^n$ following Equation (10)
    Augment the intrinsic rewards and update the policy network $\pi_\theta$ using any RL method
  **end for**

---

# 6 EXPERIMENTS

We compare the experimental performance of EMI to recent prior works on both of the low-dimensional locomotion tasks with continuous control from rllab benchmark (Duan et al., 2016) and the complex vision-based tasks with discrete control from the Arcade Learning Environment (Bellemare et al., 2013). For the locomotion tasks, we chose SwimmerGather and SparseHalfCheetah environments for direct comparison against the prior work of Fu et al. (2017). SwimmerGather is a hierarchical task where a two-link robot needs to reach green pellets, which give positive rewards, instead of red pellets, which give negative rewards. SparseHalfCheetah is a challenging locomotion task where a cheetah-like robot does not receive any rewards until it moves 5 units in one direction.

For vision-based tasks, we selected Freeway, Frostbite, Venture, Montezuma's Revenge, Gravitar, and Solaris for comparison with recent prior works (Pathak et al., 2017; Fu et al., 2017). These six Atari environments feature very sparse reward feedback and often contain many moving distractor objects which can be challenging for the methods that rely on explicit decoding of the full observations (Oh et al., 2015).

## 6.1 IMPLEMENTATION DETAILS

We use TRPO (Schulman et al., 2015) for policy optimization because of its capability to support both the discrete and continuous actions and its robustness with respect to the hyperparameters. In the locomotion experiments, we use a 2-layer fully connected neural network as the policy network. In the Atari experiments, we use a 2-layer convolutional neural network followed by a single layer fully connected neural network. We convert the 84 x 84 input RGB frames to grayscale images and resize them to 52 x 52 images following the practice in Tang et al. (2017). The embedding dimensionality is set to $d = 2$ in all of the environments except for Gravitar and Solaris where we set $d = 8$ due to their complex environment dynamics. We use Adam (Kingma & Ba, 2015) optimizer to train embedding networks. Please refer to Appendix A.1 for more details.

## 6.2 LOCOMOTION TASKS WITH CONTINUOUS CONTROL

We compare EMI with TRPO (Schulman et al., 2015), EX2 (Fu et al., 2017), and ICM (Pathak et al., 2017) on two challenging locomotion environments: SwimmerGather and SparseHalfCheetah. Figure 4 shows that EMI outperforms the baseline methods on both tasks. Figure 3b visualizes the scatter plot of the learned state embeddings and an example trajectory for the SparseHalfCheetah experiment. The figure shows that the learned representation successfully preserves the similarity in observation space. Please refer to Appendix A.3 for further experiments including ablation study.

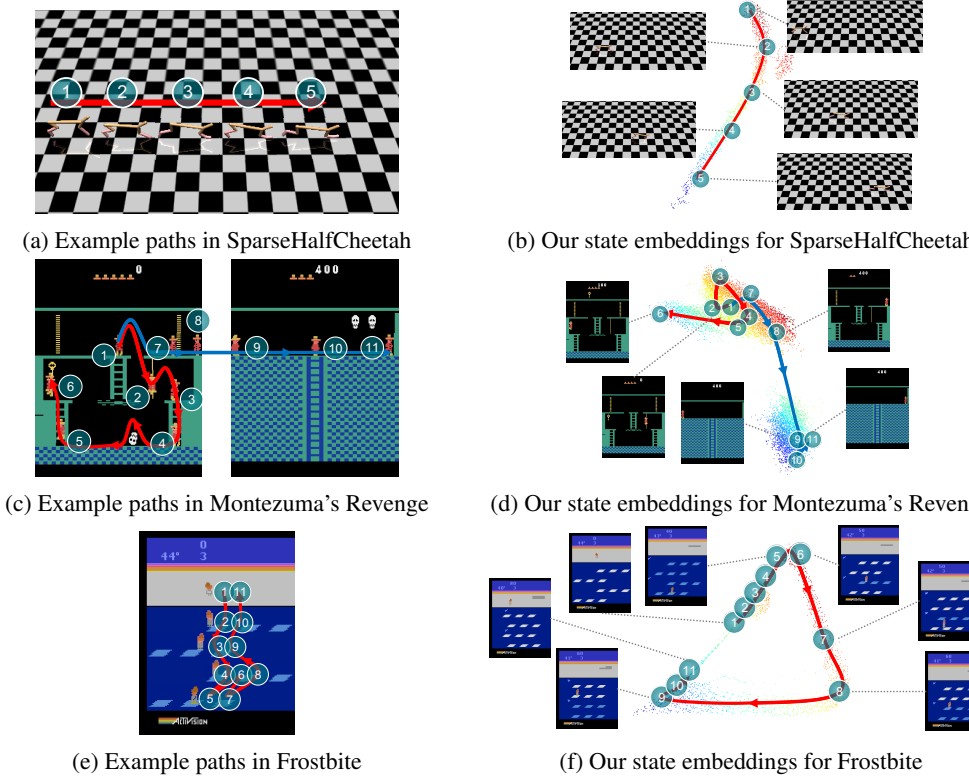

(a) Example paths in SparseHalfCheetah (b) Our state embeddings for SparseHalfCheetah

(c) Example paths in Montezuma's Revenge (d) Our state embeddings for Montezuma's Revenge

(e) Example paths in Frostbite (f) Our state embeddings for Frostbite

Figure 3: Example sample paths in our learned embedding representations. Note the embedding dimensionality $d$ is 2, and thus we did not use any dimensionality reduction techniques.

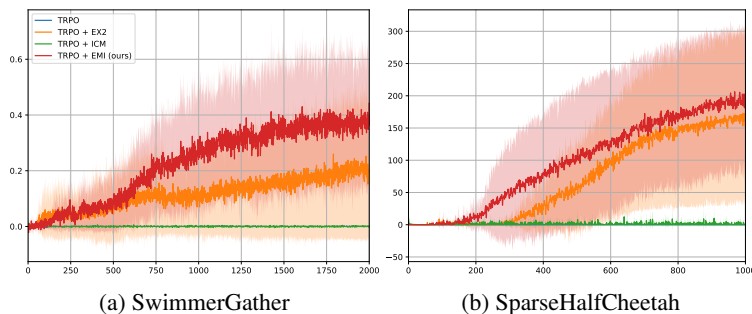

(a) SwimmerGather (b) SparseHalfCheetah

Figure 4: Performance of EMI on locomotion tasks with sparse rewards compared to baseline methods (TRPO, EX2, ICM). The solid line is the mean reward (y-axis) of 5 different seeds at each iteration (x-axis) and the shaded area represents one standard deviation from the mean.

## 6.3 VISION-BASED TASKS WITH DISCRETE CONTROL

For vision-based exploration tasks, our results in Figure 5 show that EMI achieves the state of the art performance on Freeway, Frostbite, Venture, and Montezuma's Revenge in comparison to the baseline exploration methods. Figures 3c to 3f illustrate our learned state embeddings $\phi$. Since our embedding dimensionality is set to $d = 2$, we directly visualize the scatter plot of the embedding representation in 2D. Figure 3d shows that the embedding space naturally separates state samples into two clusters each of which corresponds to different rooms in Montezuma's revenge. Figure 3f shows smooth sample transitions along the embedding space in Frostbite where functionally similar states are close together and distinct states are far apart. For information about how our error term $S(s, a)$ works in those vision-based tasks, please refer to Appendix A.2.

Extending our experiments in Figure 4 and Figure 5, we further compare EMI with other exploration methods as shown in Table 1. EMI shows the outstanding performance on 6 out of 8 environments.

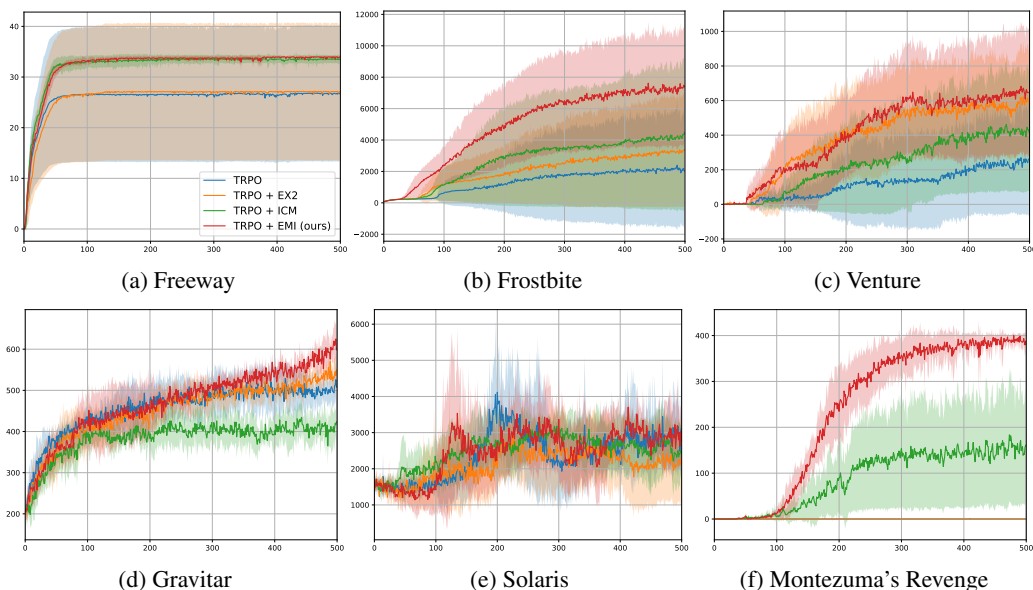

(a) Freeway  (b) Frostbite  (c) Venture

(d) Gravitar  (e) Solaris  (f) Montezuma's Revenge

Figure 5: Performance of EMI on sparse reward Atari environments compared to the baseline methods (TRPO, EX2, ICM). EMI in (a), (b), (d), (e) uses relative diversity intrinsic rewards. Prediction error intrinsic rewards are used in (c), (f). The solid line is the mean reward (y-axis) of 5 different seeds at each iteration (x-axis) and the shaded area represents one standard deviation from the mean.

| | EMI (5 seeds) | EX2 (5 seeds) | ICM (5 seeds) | SimHash | VIME | TRPO (5 seeds) |
|---|---|---|---|---|---|---|
| SwimmerGather | **0.442** | 0.200 | 0 | 0.258 | 0.196 | 0 |
| SparseHalfCheetah | **194.9** | 153.7 | 1.4 | 0.5 | 98.0 | 0 |
| Freeway | **34.0** | 27.1 | 33.6 | 33.5 | - | 26.7 |
| Frostbite | **7388** | 3387 | 4465 | 5214 | - | 2034 |
| Venture | **646** | 589 | 418 | 616 | - | 263 |
| Gravitar | 599 | 550 | 424 | **604** | - | 508 |
| Solaris | 2775 | 2276 | 2453 | **4467** | - | 3101 |
| Montezuma | **387** | 0 | 161 | 238 | - | 0 |

Table 1: Mean score comparison on baseline methods. We compare EMI with EX2 (Fu et al., 2017), ICM (Pathak et al., 2017), SimHash (Tang et al., 2017), VIME (Houthooft et al., 2016) and TRPO (Schulman et al., 2015). EX2, ICM and TRPO columns are average of 5 seeds runs coherent to Figure 4 and Figure 5. SimHash and VIME results are reported in previous works. All exploration methods here are implemented based-on TRPO policy. Results of SparseHalfCheetah and SwimmerGather are reported around 5M and 100M time steps respectively. Results of Atari environments are reported around 50M time steps.

## 7 CONCLUSION

We presented EMI, a practical exploration method that does not rely on direct generation of high dimensional observations while extracting the predictive signal that can be used for exploration within a compact representation space. Our results on challenging robotic locomotion tasks with continuous actions and high dimensional image-based games with sparse rewards show that our approach transfers to a wide range of tasks and shows state of the art results significantly outperforming recent prior works on exploration. As future work, we would like to explore utilizing the learned linear dynamic model for optimal planning in the embedding representation space. In particular, we would like to investigate how an optimal trajectory from a state to a given goal in the embedding space under the linear representation topology translates to the optimal trajectory in the observation space under complex dynamical systems.

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

# A  APPENDIX

## A.1  EXPERIMENT HYPERPARAMETERS

In all experiments, we use Adam optimizer with a learning rate of 0.001 and a minibatch size of 512 for 3 epochs to optimize embedding networks. In each iteration, we utilized collected TRPO batch at each iteration to train embedding networks except for SparseHalfCheetah which uses FIFO replay buffer of size 250000. The embedding dimensionality is set to $d = 2$ in all of the environments except for Gravitar and Solaris where we set $d = 8$. Relative diversity term is used as an intrinsic reward with the weight of 0.1, except for Venture and Montezuma's Revenge where the intrinsic reward is set as a prediction error term with the weight of 0.001. The following tables give the detailed information of the remaining hyperparameters.

| Environments | SwimmerGather | SparseHalfCheetah |
|---|---|---|
| TRPO method | Single Path | |
| TRPO step size | 0.01 | |
| TRPO batch size | 50k | 5k |
| Policy network | A 2-layer FC with (64, 32) hidden units (tanh) | |
| Baseline network | A 32 hidden units FC (ReLU) | Linear baseline |
| $\phi$ network | Same structure as policy network | |
| $\psi$ network | A 64 hidden units FC (ReLU) | |
| Information network | A 2-layer FC with (64, 64) hidden units (ReLU) | |
| Error network | State input passes the same network structure as policy network. Concat layer concatenates state output and action. A 256 units FC (ReLU) | |
| Max path length | 500 | |
| Discount factor | 0.995 | |
| $\lambda_{\text{info}}$ | 0.05 | |
| $\lambda_{\text{sparsity}}$ | 10000 | |
| $\lambda_{\text{unitKL}}$ | 0.1 | |

Table 2: Hyperparameters for MuJoCo experiments.

| Environments | Freeway, Frostbite, Venture, Montezuma's Revenge, Gravitar, Solaris |
|---|---|
| TRPO method | Single Path |
| TRPO step size | 0.01 |
| TRPO batch size | 100k |
| Policy network | 2 convolutional layers (16 8x8 filters of stride 4, 32 4x4 filters of stride 2), followed by a 256 hidden units FC (ReLU) |
| Baseline network | Same structure as policy network |
| $\phi$ network | Same structure as policy network |
| $\psi$ network | A 64 hidden units FC (ReLU) |
| Information network | A 2-layer FC with (64, 64) hidden units (ReLU) |
| Error network | State input passes the same network structure as policy network. Concat layer concatenates state output and action. A 256 units FC (ReLU) |
| Max path length | 4500 |
| Discount factor | 0.995 |
| $\lambda_{\text{info}}$ | 0.1 |
| $\lambda_{\text{sparsity}}$ | 100 |
| $\lambda_{\text{unitKL}}$ | 0.5 |

Table 3: Hyperparameters for Atari experiments.

## A.2 EXPERIMENTAL EVALUATION OF THE ERROR MODEL

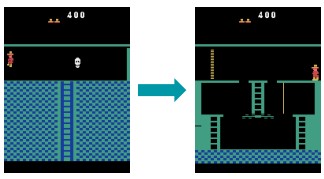 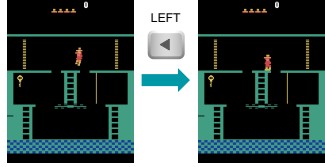 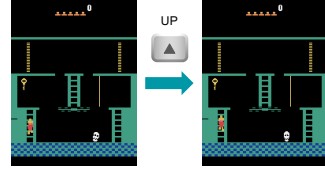

(a) $s_t$ and $s'_t$ are from different rooms with distant background images.

(b) The agent is already off the platform in $s_t$.

(c) The agent climbs up the ladder as expected.

Figure 6: Example transitions that entail large or small instances of the error term $S(s, a)$, in Montezuma's Revenge.

In order to understand how the error term $S(s, a)$ in EMI works in practice, we visualize three representative transition samples in Figure 6 and check the residual error norm without the error term ($\|\phi(s'_t) - (\phi(s_t) + \psi(a_t))\|_2$) and the error term norm ($\|S(s_t, a_t)\|_2$).

In the case of Figure 6a, due to the discrepancy between the two different background images, $\|\phi(s_t) - \phi(s'_t)\|_2$ becomes large which makes the residual error as well as the error term larger, too. For this specific sample, the residual error norm without the error term was 2.72 and the norm of the error term was 0.0296. Figure 6b describes the case where the action chosen by the policy has no effect on $s'_t$ i.e. $P(s'_t|s_t, a_t) = P(s'_t|s_t)$. Linear models without any noise terms can easily fail in such events. Thus, the error term in our model gets bigger to mitigate the modeling error. The norm of the residual error without the error model for this example transition was 3.81, and its error term had a norm of 0.0309.

On the other hand, Figure 6c represents cases that the chosen action works in the environment as intended. The residual error norm for this sample was 0.79 without the error term and the error term norm was 0.0082.

In conclusion, we observed the error terms generally had much larger norms in the cases such as Figure 6a (0.0296) and Figure 6b (0.0309) compared to the case like Figure 6c (0.0082), in order to alleviate the occasional irreducible large residual errors under the linear dynamics model.

## A.3 ABLATION STUDY

Figure 7 shows the ablation study of loss terms in EMI to verify the influence of each factor. Ablating a single factor like the information term, the linear dynamics with sparse noise or the unit KL divergence constraint degrades performance significantly. It means that each factor has a non-trivial impact on EMI. Also, simultaneously ablating the information gain term with another factor diminishes reward into zero. It denotes that the information gain term has the most critical impact on EMI.

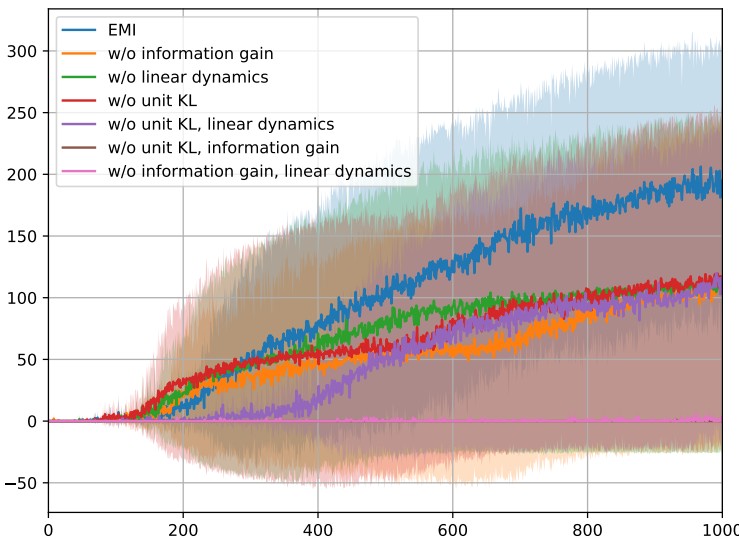

Figure 7: Ablation study of loss terms in EMI on SparseHalfCheetah environment. Each solid line represents the mean reward of 5 random seeds.

In reward augmentation process, EMI agent computes intrinsic reward $r_d$ and then learns from $r = r_{env} + \alpha r_d$. Figure 8 shows the impact of $\alpha$ in EMI. Although $\alpha = 0.1$ gives the best performance, other choices also give comparable performance. It can be concluded that EMI is robust to the choice of intrinsic reward coefficient.

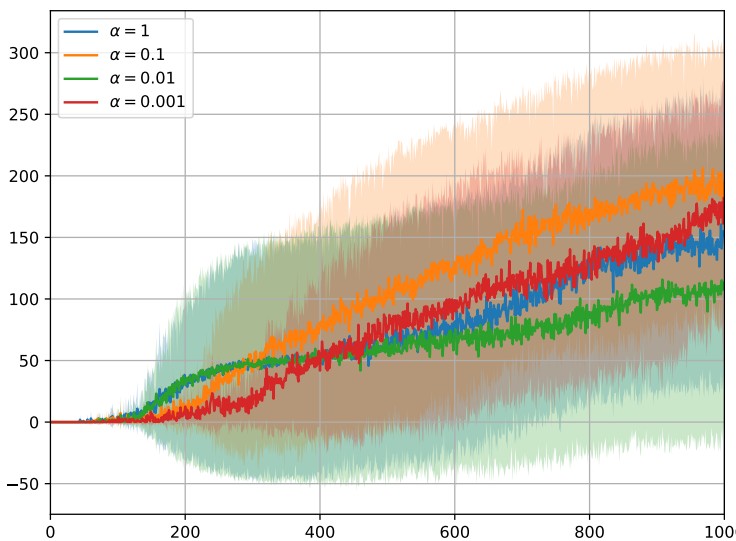

Figure 8: Study of intrinsic reward coefficient $\alpha$ in EMI on SparseHalfCheetah environment. Each solid line represents the mean reward of 5 random seeds.

