# OpenReview forum: "EMI: Exploration with Mutual Information Maximizing State and Action Embeddings"
_ICLR.cc/2019/Conference_

### Official Review · AnonReviewer1 · 2018-10-24
**Review of EMI**

**Rating:** 7
**Confidence:** 3

**Review:**

This paper introduces actions as a co-predictor of next-states and the predicted (from current and next state) in the context of (model-based) RL. In addition they incorporate the idea of using a JSD-based objective do prediction (as the Deep InfoMax paper), which is novel to RL. The enforce a linear structure between current / next states and actions with an additional sparse nonlinear term computed from both current states and actions. From this, they are able to quantify the amount of novelty in the representation space as a measure of exploration, which can be used as an intrinsic reward.

I found the paper to be very well-written and easy to understand. The prediction part is similar to that used in CPC structurally, except they include the action in two different prediction tasks and they have some built-in intrinsic rewards, which is good.

I had some issues with the motivations of some of the loss functions.
- The JSD-based objective makes sense, but I don't think it's correct to call it an "approximation" to the KL (this is only true where the log-ratio of the joint and the product of marginals is small). Rather, it would be better to describe this choice as simply using a different measure between the joint and marginals.
- It seems like the best motivation for having linear relations is you can do multiple predictions using the same state / action encodings.
- For measuring exploration (11) couldn't one just use the predictor models T? How does the output of T (perhaps correctly normalized with the marginals) correlate with (11)?

Other notes:
Page 2:
Figure 1 is awfully confusing. Could this be clarified a little bit? I’m not sure what the small dots or their colors are supposed to represent.

Could diversity also be added by adding a prior to the state representations (as is done in Deep InfoMax)?

Why were the vision experiments stopped at 500 x 100k (500 million) frames?  I can’t validate the SOTA claims, but it seems like the model is still improving: are there’s further experiments?

An ablation study would be nice comparing the different hyper parameters (intrinsic rewards, diversity, etc).

---

> ### Author Response · Authors · 2018-11-19
> **Response to Reviewer 1**
>
> Dear Reviewer,
>
> We would like to thank you for the time and effort spent providing the feedback. We address the questions below and also updated corresponding response in the pdf submission.
>
> 1. “JSD as an "approximation" to the KL”: We agree that it would be better not to state JSD as “approximation” to the KL. We modified our statement to call it “different measure” in section 4.1.
>
> 2. “Using predictor models with output of T”: This is a valid suggestion, as by the mutual information maximizing objective, the network T is encouraged to output smaller values for novel samples. We ran experiments in SparseHalfCheetah environment, and were able to confirm some degree of exploration effect of the intrinsic rewards derived by the outputs of the network T. Although the results are not comparable to our proposed method (average return of 75 vs. 200 after 1,000 iterations in SparseHalfCheetah), it will be an interesting future research direction.
>
> 3. “Figure 1 is awfully confusing”: Sorry for the confusion. We updated Figure 1 and hope it clarifies the big picture.
>
> 4. “Adding a prior to the state representations”: Regularizing the distribution of state embeddings instead causes the optimization process to be much more unstable. This is because the distribution of states is much more likely to be skewed than the distribution of actions, especially during the initial stage of optimization, so the Gaussian approximation of the distribution of state embeddings in the KL regularization term becomes much less accurate in contrast to the distribution of actions. We added this statement to section 4.2.
>
> 5. “Vision experiments stopped at 50 frames”: We stopped at 50 million frames to perform fair comparisons to other baseline methods. TRPO-based exploration methods such as EX2 [1], SimHash [2] stopped at 50 million frames.
>
> 6.” An ablation study would be nice”: We added ablation study in Appendix. In figure 7, We ablated on each loss terms of Equation 9 to show the impact of each term. In figure 8, we tested on different intrinsic reward hyper-parameters to show the robustness of our method.
>
> [1] Justin Fu, John Co-Reyes, and Sergey Levine. Ex2: Exploration with exemplar models for deep reinforcement learning. In Advances in Neural Information Processing Systems, pp. 2577–2587, 2017.
>
> [2] Haoran Tang, Rein Houthooft, Davis Foote, Adam Stooke, Xi Chen, Yan Duan, John Schulman, Filip DeTurck, and Pieter Abbeel. # exploration: A study of count-based exploration for deep reinforcement learning. In Advances in Neural Information Processing Systems, pp. 2753–2762, 2017.

---

> > ### Public Comment · (anonymous) · 2018-12-06
> > **Claims of SOTA performance on Atari are misleading**
> >
> > As an expert in this area, it troubles me to see so many exploration papers at this year's ICLR comparing against weak baselines. I'm particularly bothered by this line:
> >
> > "EMI achieves the state of the art performance on Freeway, Frostbite, Venture, and Montezuma’s Revenge in comparison to the baseline exploration methods"
> >
> > This is a misuse of terminology. State-of-the-art means "outperforms all other methods" not "some other methods". If you're going to use this term, you really ought to mention Ostrovski et al.'s well-known paper "Count-Based Exploration with Neural Density Models". Some of their configurations reach approx 2,500 points on Montezuma's Revenge by 50 million frames (even their worst reach around 1,500 points), so your algorithm is definitely not the strongest amongst *all* exploration methods on Montezuma's Revenge. Instead, it would be better if you just said "EMI outperforms the baseline exploration methods on X, Y, Z" (and justify why you have not included Ostrovski et al.'s agent in the comparison).
> >
> > If you're only comparing against TRPO agents, as you've mentioned to another public commenter, then this ought to be mentioned, because it's not clear at all from the paper. This choice seems pretty contrived though. For example, Bellemare et al.'s AC3+ agent uses a very similar underlying training algorithm. Claiming SOTA for TRPO while ignoring A3C (which is generally a weaker algorithm) feels like splitting hairs.
> >
> > One further thing that troubles me is that you've tuned the algorithm for individual games. As per the appendix:
> > "The embedding dimensionality is set to d =  2 in all of the environments except for Gravitar and Solaris where we set d = 8.  Relative diversity term is used as an intrinsic reward with the weight of 0.1, except for Venture and Montezuma’s Revenge where the intrinsic reward is set as a prediction error term with the weight of 0.001."
> >
> > This is very non-standard practice in Atari, and makes your comparisons against agents that do not use per-game configurations kind of meaningless.

---

> > > ### Author Response · Authors · 2018-12-12
> > > **Response to Public Comment 3**
> > >
> > > Thank you very much for your thoughtful comments. We address your questions below.
> > >
> > > - Intrinsic reward setting:
> > > First, we want to make it clear that we *do not* tune the hyperparameters separately for each game. Although we used different intrinsic reward functions, we did not tune the hyperparameters independently for all environments. We use 0.1 for all games with relative diversity intrinsic reward and use 0.001 for all games with prediction error intrinsic reward. Therefore, the concern that we perform per-game tuning is simply incorrect. Regarding the embedding dimension, however, we did use a different embedding dimension for Gravitar and Solaris only, which did provide a small improvement, but we will rerun this experiment with an embedding dimensional of 2 for the final such that the results here are fully consistent.
> > >
> > > - "State of the art" misleading
> > > We appreciate the point about claims regarding state-of-the-art results. We will remove this claim from the paper, and replace it with a statement that our method outperforms the methods that we compare to in our experiments, which are set up for a head-to-head comparison against prior exploration methods while using the same base RL algorithm. We would be happy to revise these claims further based on what the reviewers and area chair might suggest.
> > >
> > > - TRPO vs A3C:
> > > We observed that DQN and A3C are very sensitive to changes in hyperparameters (figure 9 in Mnih et al. (2016)) making the comparison difficult. We decided to use policy gradient method (TRPO) because of the stability and speed. This choice also allowed us to easily test our method on continuous control tasks on Mujoco environment which neither Ostrovski et al. nor Bellemare et al.'s work reports the results on. This is made possible because we don't restrict the method to be specialized for image states (pixel-CNN density model) and because we decided to use TRPO. Having said that, with regards to Montezuma's Revenge, TRPO as a policy gradient based method, is not particularly suited for this type of task. This is supported [from the response in #Exploration] by the related method A3C, in which the pseudo-count method in Bellemare et al. (2016) also leads to only minor improvements (A3C+), despite Bellemare et al. (2016) achieving state-of-the-art with DQN.
> > >
> > > - Bellemare et al. uses a very similar underlying training algorithm:
> > > You seem to have misunderstood our training algorithm. The only similarity is that Bellemare et al. motivates the approach with information gain. The information gain in Bellemare et al. is defined on distributions of pixels in the density model of raw pixels. Our approach on the other hand is to define the information gain on both the embedding representation of the states (\phi) and the representation of actions (\psi) and maximize the lower bound (from f-GAN) of the mutual information between I([\phi(s), \psi(a)]; \phi(s')) and I([\phi(s), \phi(s')]; \psi(a)). This part of the proposed algorithm is much more similar to the ideas in ICM and much less Bellemare et al. Hence, we implemented ICM on TRPO and included it as a baseline. Another distinction is that, both ICM and EMI aim to learn the generic embedding representations not restricted to 2D image states. This allows us to effortlessly adopt the method for continuous control tasks (see Fig 4).
> > >
> > > - Sequential generative decoding methods (Ostrovski et al. Bellemar et al.):
> > > Both the approaches require sequential generative models and must model the distribution over state observations (pixel's color value) pixel by pixel, which can make them difficult to scale. On top of that, A3C implementation of Ostrovski et al. maintains 16 copies of density models during training. While the density based approaches are promising methods for building advanced exploration strategies, they require much more effort to tune and stabilize, and can be much slower to run. On the other hand, the proposed and the baseline methods (EMI, EX2, ICM) do not require the generative decoding of the original observation space. We will make this distinction clear in the paper.

---

### Official Review · AnonReviewer2 · 2018-11-01
**Review of EMI**

**Rating:** 7
**Confidence:** 4

**Review:**

This is a very interesting paper about a novel approach to exploration in agents with state and action representations, making heavy use of recent progress in the use of deep learning for estimating and maximizing mutual information, as well as introducing an approach to model the latent space dynamics with a linear models with sparse errors.

A closely related work which is not mentioned is the work of Thomas et al 2017 arXiv:1708.01289 where they also maximize mutual information between distributed representations of actions (policies, actually) and of distributed representations of changes in the state (as the result of applying the policy).

The phrase 'functionally similar states' is used several times and would require a bit of explanation.

I would also like to see more motivations for the two different reward functions r_e and r_d, and why one should be computed before the update while the other should be computed after.

Regarding the experiments, and this is probably the weakest part of this paper, I would have expected to see comparisons against several of the numerous exploration methods which have been proposed in the past and are discussed in the paper (with many negative comments about their weakness, but no empirical support provided). Only one (EX2) was compared. The comparison with TRPO is without exploration (if I understand well, but should be stated clearly).  It's also not clear how these results compare to the best reported results on these games (whether or not exploration is used).

---

> ### Author Response · Authors · 2018-11-19
> **Response to Reviewer 2**
>
> Dear Reviewer,
>
> We would like to thank you for the time and effort spent providing the feedback. We address the questions below and also updated corresponding response in the pdf submission.
>
> 1. “Mention the work of Thomas et al 2017”: Thank you for spotting this. We added the work to the related works section.
>
> 2. “Motivations for the two different reward functions r_e and r_d, and updating order”: We propose two different intrinsic reward functions to show that EMI can be utilized with diverse intrinsic reward functions and is not constrained to the particular form of intrinsic reward function. As RL agents behave differently with different intrinsic reward functions in various environments, we can use EMI with appropriate intrinsic reward functions depending on environments. (Example: 1) Diversity reward gives good performance in SparseHalfCheetah (Mujuco) where the task is to go as far away from the origin. 2) Prediction error reward works well in Montezuma’s Revenge (Atari) where the agent can receive high prediction error by entering different rooms.)
>
> 3. “Comparisons against several of the numerous exploration methods”: We had experiments for both EX2 [1] and ICM [2] on Atari experiments. But for locomotion tasks, we only compared against EX2. This was because ICM was mainly designed for discrete actions. To this end, we extended ICM to accommodate continuous actions (by replacing the cross entropy loss for categorical policy with L2 loss for continuous policy) ran it on continuous locomotion tasks to as another baseline in the experiments section. Also, we added Table 1 to further compare the final performance of EMI against other exploration methods.
>
> [1] Justin Fu, John Co-Reyes, and Sergey Levine. Ex2: Exploration with exemplar models for deep reinforcement learning. In Advances in Neural Information Processing Systems, pp. 2577–2587, 2017.
>
> [2] Deepak Pathak, Pulkit Agrawal, Alexei A Efros, and Trevor Darrell. Curiosity-driven exploration by self-supervised prediction. In International Conference on Machine Learning, volume 2017, 2017.

---

### Official Review · AnonReviewer4 · 2018-11-15
**Interesting paper about intrinsic rewards for exploration, via embeddings which improve mutual information**

**Rating:** 5
**Confidence:** 4

**Review:**

The paper proposes an approach for exploration via reward bonuses based on a form of surprise. The surprise factor is based on the next state of a particular transition, and the error in the embedding space to satisfy a linear dynamics formulation. The embedding space of the states and actions are optimized to increase the mutual-information in predicting next state, and current action - encouraging meaningful embeddings with more training, and hence gradual fading away of the extrinsic rewards.

The paper is mostly well-written, and the idea is interesting. The experimental results do show that the proposed reward augmentation leads to better performing policies, but the claims in the experimental section need to be less strong ("outperforms the baseline by a large margin" - Figure 4 - overlapping error bars; "state of the art" - Figure 5 - again, error bars, and no improvement in some domains.) But overall, I think the paper can be accepted as it is an interesting approach.

Below are some comments that I hope the authors address in their rebuttal, followed by some possible typos in the current draft.

- Theorem 1 content placement: The organization here is rather unclear. Currently, you present Theorem 1, and then talk about using "JSD instead of MI". Maybe this is a last minute mistake. In either case, it is strongly suggested that the section be reworked to be clearer.

- JSD is upper bounded by ln(2); your bounds (7) and (8) would change consequently too.

- Training regime employed:
  3 epochs-512 minibatch -> assuming distinct minibatches are sampled, (512x3) samples used
  Collected (5k*500; sparsehalfcheetah)/(50*500; swimmercatcher)/(100k*4500; atari)
  Is this the sample usage for training? Axis labels for all plots are missing -- specifically scale of x-axis.
  Commenting on the sample complexity -- especially as the embedding network seems easy-to-train (or insufficiently trained), would be good; optimizing a lower-bound insufficiently leads one to doubt if the bound is meaningful at all. Is the huge batch of samples mostly used in TRPO/RL part of the infrastructure?

- Discussing extrinsic rewards: the pros. vs. cons of the two reward formulations, why both are used etc. would be useful.

- Embedding dimension: d=8 in Gravitar and Solaris, but performance is less significant (no significance) in these domains. Is this due to insufficient training?

- RL method: Including details about form of TRPO used in appendix would be good (vine/single-path). Further if entropy regularization is used, how does the exploration interplay work.

- A.2 is an interesting section. A linear dynamics model being effective in MuJoCo tasks seems plausible. But an Atari example is definitely more interesting. Therefore this section can be clearer - specifically distinction between residual error and sample error.

Typos:
- Appendix \lambda parameters unclear
- Appendix step-size information contradictory.


Post-response comment: while I do think the approach is interesting, the utility of it is mostly demonstrated currently through empirical experiments. These experiments are preliminary, but used to make strong arguments for the effectiveness of the proposed approach. Further, upon highlighting this in my review, the authors disagree and think it’s empirical validity is rather superior. This leaves me concerned, and after thinking about it further, I do not think this is sufficient for acceptance. Therefore, I’m reducing my score to a 5.

PS: the characterization of irreducible error as a product of the limitation of a linear model may be inaccurate.

---

> ### Author Response · Authors · 2018-11-19
> **Response to Reviewer 4**
>
> Dear Reviewer,
>
> We would like to thank you for the time and effort spent providing the feedback. We address the questions below and also updated corresponding response in the pdf submission.
>
> 1. “Theorem 1 organization”: We reorganized the structure around Theorem 1 in the paper.
>
> 2. “JSD is upper bounded by ln(2)”: In f-GAN [1], they actually derived the lower bound of D_{JSD} = D_{KL} (P||M)+D_{KL} (Q||M), instead of D_{JSD} =1/2(D_{KL} (P||M)+D_{KL} (Q||M)). Thus our D_{JSD} formulation coherent with [1] is upper-bounded by log(4). As there was a typo in the upper-bound of D_{JSD}, we fixed it from log(2) to log(4).
>
> 3. “Sample usage for training and sample complexity”: At each iteration in the atari environment, 100k (state, action, reward) pairs are sampled as a batch. The number 4500 is the max path length, which means that the length of each episode should not exceed 4500. (When the 4500th state is reached, the game is reset even though the agent does not reach the terminal state.) Thus, if we assume that every episode is reset at 4500th state then we have about 100k / 4500 = 22 episodes in the batch at each iteration. After the batch of 100k samples is collected, we split the batch into minibatches of size 512 and train our network. (We repeat this minibatch training 3 times because epoch is 3.)
> For the plot axes, the x-axis is iteration and the y-axis is the mean reward. We clarified the axes in the figure captions.
> For the sample complexity, the huge batch is used to train both TRPO policy network and EMI embedding network at approximately the equal amount. It is not the case that the embedding network underfits.
>
> 4. “Pros. vs. cons of the two reward formulations”: We propose two different intrinsic reward functions to show that EMI can be utilized with diverse intrinsic reward functions and is not constrained to the certain intrinsic reward function. As RL agents behave differently with different intrinsic reward functions in various environments, we can use EMI with appropriate intrinsic reward functions depending on environments. (Example: 1) Diversity reward gives good performance in SparseHalfCheetah (Mujoco) where the agent should go as far from the origin. 2) Prediction error reward works well in Montezuma’s Revenge (Atari) where the agent can receive high prediction error by entering different rooms.)
>
> 5. “Performance is less significant in Gravitar and Solaris with d=8”: We used higher dimension embedding for Gravitar and Solaris because their environment dynamics are much challenging to predict with d=2 linear embeddings. Games like Freeway and Montezuma’s Revenge give discrete actions related to 4-way direction moves in 2D space. However, Gravitar and Solaris have different kinds of action dynamics and consist of visually more complex states. These factors make the performance of Gravitar and Solaris to be less significant compared to other games.
>
> 6. “RL method: details about TRPO and entropy regularization”: We added the details about TRPO in appendix. (We use Single path method for TRPO.) The entropy regularization is a technique that is widely used for policy-based RL algorithms. We think that it would be possible to combine these two as entropy regularization seeks to diversify sampled actions and EMI seeks to diversify sampled states.
>
> 7. “Section A.2 can be clearer”: We polished Appendix 2 in terms of the distinction between residual errors and sample errors (error terms).
>
> 8. “Claims to be less strong”: As RL problems exhibit high variance on the return especially on environments with sparse rewards, we do not agree that one method is not better than the other just because of the overlapping error bars. Our experiments are all performed in sparse reward setting and the return plots in Figure 4 and a subset of Figure 5 show clear advantage of our method over other exploration baselines. Having said that, we moderated the claim a bit.
>
> 9. “Typos: lambda, step size”: Thank you for pointing this out. We modified Equation 9 to make it coherent with lambdas in Appendix. TRPO step size is hyper-parameter related to KL divergence in TRPO algorithm which uses conjugate gradients to update policy network.
>
> [1]  Sebastian Nowozin, Botond Cseke, and Ryota Tomioka. f-gan: Training generative neural samplers using variational divergence minimization. In Advances in Neural Information Processing Systems, pp. 271–279, 2016.

---

### Public Comment · (anonymous) · 2018-10-01
**Interesting paper.**

You combine 1. an idea for learning representations and 2 . an idea for factorizing the dynamics into a linear term plus a sparse irreducible error term.

It would be informative to ablate these. As it stands we have no information as to which was important.

In addition you should not claim these results are state of the art, there are much better results on montezuma's revenge for example.

---

> ### Author Response · Authors · 2018-11-19
> **Response to Public Comment 1**
>
> Thank you very much for your thoughtful comments. We address your questions below.
>
> We added the ablation study section in Appendix. In figure 7, We ablated on each loss terms of Equation 9 to show the impact of each term. Also, we agree that there are other great results in Atari domains, but in our paper, we claim SOTA among TRPO-based exploration methods.

---

> > ### Public Comment · (anonymous) · 2018-11-21
> > **Thank you**
> >
> > Thanks for these ablations!

---

### Public Comment · (anonymous) · 2018-10-30
**Question**

I enjoyed your paper, and I have a question.

At the beginning of page 6, you mentioned that 'the relative diversity term should be computed after the representations are updated ... while the prediction error term should be computed before the update.'

Could you explain the reason why the prediction error term should be computed before the update in detail?
For example, what if we calculate the prediction error term after the update, too?

I think readers may wonder the clear reason for this. :)

---

> ### Author Response · Authors · 2018-11-19
> **Response to Public Comment 2**
>
> Thank you very much for your thoughtful comments. We address your questions below.
>
> The relative diversity term is computed after the update because we desire to give high intrinsic rewards to states that are far away from common states in learned embedding space. However, the prediction error term should be computed before the update because we desire to give high intrinsic rewards to states that give more surprise (high prediction error). If we compute the prediction error term after the update, then novel states would not give high intrinsic reward because the update reduces prediction error already.

---

### Meta-Review · Area_Chair1 · 2018-12-14
**Some interesting ideas with concerns about motivation and experiments**

**Confidence:** 4
**Recommendation:** Reject

**Metareview:**

This paper proposes a method to compute embeddings of states and actions that facilitate computing measures of surprise for intrinsic reward. Though some of the ideas are quite interesting, there are currently issues with the experiments and the motivation.

The experiments have high variance across the 5 runs, with significant overlap of shaded regions representing just one standard deviation from the mean. It is hard to draw any conclusions about improved performance, and statements like the following are much too strong: "For vision-based exploration tasks, our results in Figure 5 show that EMI achieves the state of the art performance on Freeway, Frostbite, Venture, and Montezuma’s Revenge in comparison to the baseline exploration methods." Further, the proposed approach has three new hyperparameters (lambdas), without much understanding into how to set them or their effect on the results. Specific values are reported for the different game types, without explanation for how or why these values were chosen.

Similarly strong claims, that are not well substantiated, are given for the proposed approach. This paper seems to suggest that this is a principled approach to using surprise for exploration, contrasted to other ad-hoc approaches ("Other approaches utilize more ad-hoc measures (Pathak et al., 2017; Tang et al., 2017) that aim to approximate surprise."). Yet, the paper does not define surprise (say by citing work by Itti and Baldi on Bayesian surprise), and then proposes what is largely a intuitive approach to providing a good intrinsic reward related to surprise. For example, "we show that imposing linear topology on the learned embedding representation space (such that the transitions are linear), thereby offloading most of the modeling burden onto the embedding function itself, provides an essential informative measure of surprise when visiting novel states." This might be intuitively true, but I do not see a clear demonstration in Section 4.2 actually showing that this restriction provides a measure of surprise. Additionally, some of the choices in Section 4.2 are about estimating "irreducible error under the linear dynamics model", but irreducible error is about inherent uncertainty (due to stochasticity and partial observability), not due to the choice of modeling class. In general, many intuitive choices in the algorithm need to be better justified, and some claims disparaging other work for being ad-hoc should be toned down.

Overall, this paper is as yet a bit preliminary, in terms of clarity and experiments. In a further iteration, with some improvements, it could be a useful contribution for exploration in image-based environments.